# Casomorphins and Gliadorphins Have Diverse Systemic Effects Spanning Gut, Brain and Internal Organs

**DOI:** 10.3390/ijerph18157911

**Published:** 2021-07-26

**Authors:** Keith Bernard Woodford

**Affiliations:** Agri-Food Systems, Lincoln University, Lincoln 7674, New Zealand; keith.woodford@lincoln.ac.nz

**Keywords:** food-derived opioids, casomorphin, gliadorphin, opioid receptors, A1 beta-casein, beta-casomorphin-7, gut-to-brain, microbiome, DPP4

## Abstract

Food-derived opioid peptides include digestive products derived from cereal and dairy diets. If these opioid peptides breach the intestinal barrier, typically linked to permeability and constrained biosynthesis of dipeptidyl peptidase-4 (DPP4), they can attach to opioid receptors. The widespread presence of opioid receptors spanning gut, brain, and internal organs is fundamental to the diverse and systemic effects of food-derived opioids, with effects being evidential across many health conditions. However, manifestation delays following low-intensity long-term exposure create major challenges for clinical trials. Accordingly, it has been easiest to demonstrate causal relationships in digestion-based research where some impacts occur rapidly. Within this environment, the role of the microbiome is evidential but challenging to further elucidate, with microbiome effects ranging across gut-condition indicators and modulators, and potentially as systemic causal factors. Elucidation requires a systemic framework that acknowledges that public-health effects of food-derived opioids are complex with varying genetic susceptibility and confounding factors, together with system-wide interactions and feedbacks. The specific role of the microbiome within this puzzle remains a medical frontier. The easiest albeit challenging nutritional strategy to modify risk is reduced intake of foods containing embedded opioids. In future, constituent modification within specific foods to reduce embedded opioids may become feasible.

## 1. Introduction

This paper reviews and integrates evidence relating to food-derived opioid peptides in public health, focusing on casomorphins from dairy and gliadorphin peptides from cereals. The systemic nature of the evidence, which spans the gut, brain, and many internal organs, arises as a direct consequence of the widespread presence of opioid receptors throughout the human body. Influencing factors beyond diet itself include human genetic variability, specific microbiota, and aspects of health that mediate absorption of the peptides from the gut into the circulatory system. Some of these same factors then impact on inflammatory and autoimmune responses. The role of the gut and associated microbiome is clearly important within the system but much remains to be elucidated [1].

One of the challenges of investigating and documenting the wide-ranging effects of food-derived opioids is that dietary exposure is long-term, with effects often due to chronic rather than short-term high-intensity exposure, and therefore difficult to investigate within clinical trials. A further investigational challenge is that the multiple influencing factors, including genetic factors and other disease factors, in combination with the diverse locations of opioid receptors within human tissues, can lead to great effect-diversity between individuals.

The specific focus of this paper on casomorphin and gliadin peptides reflects that these are the most researched of the food-derived opioids based on the prevalence of dairy and wheat within human diets, together with evidence that they are the peptide groupings most clearly implicated in food-derived opioid health issues. However, they are not the only opioid peptides in either gluten or dairy. For example, there are opioid peptide sequences within glutenin, the alcohol-insoluble proteins that along with the alcohol-soluble gliadins comprise the overarching gluten grouping. Additionally, opioid peptides can be released from other food products. For example, barley and rye have homologous proteins to the gluten proteins found in wheat. These barley and rye proteins release opioid hordein peptides and opioid secalin peptides, respectively [2]. Indeed, there is diversity of practice within the literature as to whether the term ‘gluten’ encompasses all of the prolamin (high proline) proteins derived from species of the Triticeae, thereby including the various wheat species plus barley, rye, and triticale, or whether it should be reserved as a term for prolamin proteins from wheat. In this paper, wherever the terms ‘gluten’ and ‘gliadin’ are used, it is as encompassing terms that include relevant prolamin proteins within all of the Triticeae, but recognizing that there will be differences between species and even between strains and varieties of a species in terms of both gluten and gliadin intensity plus specific opioid structure [3]. A notable feature of gluten proteins is the variety and complexity of structure [4].

Non-homologous opioid peptides can also be released from soy as soymorphins [5] and from spinach as rubiscolins [6]. Those issues lie specifically beyond the scope of this paper.

Both dairy casomorphins and gluten peptides are largely specific to modern diets. This is because neither dairy nor cereals were of dietary importance prior to the gradual emergence of agricultural and animal domestication activities. This began to occur approximately 10,000 years ago within Neolithic communities in the Fertile Crescent of West Asia. In many parts of the world, dairy plus wheat and other gluten-producing cereals from the Triticeae have only become important dietary components in the most recent centuries. Further, there is evidence of considerably lower levels of gluten peptides in early strains of wheat compared to modern wheat varieties [7] and also in durum varieties used in pasta compared to bread-making varieties [8].

The potential importance of casomorphin and gluten peptides in relation to human health remains an emergent field. For example, the presence of opioid casomorphins in dairy was not identified until 1979 [9], and the full extent of casomorphin-relevance to multiple issues of human health is far from resolved. In contrast, early insights that coeliac disease was protein-related and with particular relevance to wheat protein were known with clarity by 1950 and suspected much earlier [10]. However, the opioid connection appears to have not been explicitly identified until 1987 [11]. The broadening of the association between coeliac disease to other gluten-producing species within the Triticeae came considerably later [3]. In regard to non-celiac aspects of gluten science and pharmacology, there is still much to be elucidated.

Emergent evidence includes that not all effects of opioid peptides are necessarily dependent on attachment to opioid receptors. For example, there is compelling evidence as to the role of toll-like receptors, and in particular TLR4, in relation to both food-derived exorphins and pharmaceutical opioids [12,13]. There is also evidence that the casomorphins directly influence the serotonergic system independent of opioid receptors [14].

In this paper, evidence will be documented to show that casomorphins and specific gliadin peptides have structural elements in common and that they thereby have potential to ‘hunt together’ in terms of inflammatory and autoimmune effects. This is reflected in arguments in favor of diets that are free of both gluten and casein (GFCF). However, there are also obvious needs to consider gluten and casein separately, given both the known structural differences and the fact that they may favor different opioid receptors. With casomorphins, it is clear that they predominantly associate with mu-opioid (MOP) receptors, whereas with gluten peptides, there is evidence that there can also be close associations with delta-opioid (DOP) receptors [15].

The key amino acid sequence at the N-terminus that is common to casomorphins and gliadorphins, and which is fundamental to the opioid characteristics, is tyrosine–proline. However, despite the widespread presence of this sequence in relation to opioid structure, together with its relevance to particular opioid characteristics, it is not a necessary structure for all food opioids [16]. The other key characteristic of casomorphins and gliadiorphins is that they are proline-rich, thereby creating resistance to peptidase enzymes.

## 2. Methodology

This is a perspectives paper that draws together and thereby integrates evidence relating to systemic effects of casomorphins and gliadorphins, spanning gut, brain, and internal organs. Accordingly, the literature was searched via Google Scholar and PubMed using various combinations of the following keywords: food-derived opioids, beta-casein, casomorphin, gliadorphin, microbiome, microbiota, opioid receptors, beta-casomorphin-7, BCM7, beta-casomorphin-9, gut-to-brain, and various specific internal organs. This literature was then filtered by the author based on manuscript focus. Some background industry information was drawn from the author’s professional involvement in agrifood systems spanning farming practices through to human nutrition.

## 3. The Role of Opioid Receptors

The widespread presence of opioid receptors spanning gut, brain, internal, and peripheral organs provides a theoretical framework to explain the current evidence and associated postulates laid out in this paper relating to diverse and systemic effects of food-derived opioids. The presence of opioid receptors in the gut and brain became well established in the 1970s [17]. It was also well understood by 1980 that opioid receptors are a key component of internal messaging systems involving endorphins and enkephalins [18]. However, identification of the widespread presence of these receptors in other organs of the body came later [19] and has been an emerging field, linked primarily to identifying and explaining the effects of opioid drugs. There is also now a substantial literature on specific molecular functional mechanisms of the opioid system [20]. This overarching suite of opioid knowledge has been central to understanding the effects that opioid drugs have on a range of internal biological processes, with opioid drugs having potential to not only exhibit inflammatory effects on specific tissues but to also disrupt internal messaging systems. This system disruption then lays a theoretical foundation for immune and auto-immune responses.

Casomorphins were first identified in the late 1970s as having opioid characteristics [9], and this is recognized with the ‘morphin’ nomenclature. Similarly, opioid peptides within gliadin were clearly identified in the 1980s [11]. However, no prior evidence has been found of extant literature on the presence of opioid receptors as a key element in identifying mechanisms whereby food-derived opioids might themselves have widespread systemic effects that extend well beyond the gut and brain. These concepts will be drawn upon in subsequent sections of the paper.

## 4. Casomorphins

By definition, casomorphins can be any opioid released from casein during digestion. In practice, the human-health interest relates to casomorphins released from beta-casein, and in particular the release of bovine beta-casomorphin-7 (bBCM7) from bovine milk. The longer chain beta-casomorphin-9 (bBCM9) is also of relevance. As background, beta-caseins are present in all mammalian milk, and in bovine milk they are the second most important of the casein proteins by volume, comprising about 35% of the casein proteins and approximately 28% of total protein [21].

The amino acid structure of bBCM7 is tyrosine–proline–phenylalaline–proline–glycine–proline–isoleucine. The first three amino acids confirm that it will be opioid in character, with the first two of fundamental importance. Additionally, the presence of the three proline amino acids in close proximity ensures that bBCM7 will be resistant to internal cleavage at the C-terminus by the peptidases [22]. Accordingly, bBCM7 is normally broken down from the N-terminus, with the key requirement being the enzyme dipeptidyl peptidase-4 (DPP4) [23]. Given the resistance to enzymatic degradation from the C-terminus, shorter-chain casomorphins are of limited practical importance within in vivo settings, despite in vitro investigations showing bBCM5 to be a stronger opioid than bBCM7 [24].

Bovine beta-casein is categorized into two broad types, these being A1 and A2, with the A1-type being fundamental to the release of bBCM7 (Figure 1). This bBCM7 peptide is located at positions 59–66 of the 209 amino acids contained within the bovine beta-casein protein [21]. In A1 beta-casein, the release of bBCM7 is facilitated by the presence of histidine at position 67, with the C-terminus bond between positions 66 and 67 being readily broken by carboxyl peptidases such as elastase [25]. In contrast, in A2 beta-casein, the amino acid at position 67 is another proline, leading to preferential cleavage of the longer peptide bBCM9 and creating a major constraint to formation of bBCM7 during in vivo digestion [21]. Although bBCM9 is itself also an opioid with consequent potential pharmacologic properties, these properties are fundamentally different, and bBCM9 is considered a potential beneficial bioactive carrying both antihypertensive properties [26] and antioxidant properties [27]. Additionally, in contrast to A1 beta-casein and bBCM7, no digestive differences have been recorded when A2 beta-casein digests containing bBCM9 are tested with and without naloxone [28]. Human-based in vivo data on beta-casein metabolism to casomorphins and intermediate peptides has been summarized within a systematic review of digestive effects undertaken by Brooke-Taylor and colleagues [22].

Subsequent to the allocation of the beta-casein terminology, it has become evident from phylogenetic analyses that A2 is the original type with A1 beta-casein being the consequence of a mutation occurring in some European cattle approximately 5000 years ago, but with considerable uncertainty as to the precise time thereof [29]. The phylogenetic evidence is also clear that there have been subsequent mutations at other loci in the bovine beta-casein protein, but with these occurring at relatively lower levels, and these are generally considered as lying within the A1 and A2 families of bovine beta-casein [30]. Accordingly, the presence of any beta-casein of the A1-type in bovine milk is evidence of at least some European cow ancestry. In contrast, the beta-caseins of sheep, goats, horses, camels, yaks, buffalo, pure African cattle, pure Asian cattle, and even human milk are all classified as being exclusively of the A2 type, with no reliable evidence of exceptions even at low levels. However, some cattle classified as African or Asian types may have a small hidden proportion of European-breed ancestry due to crossbreeding within the last 200 years, leading to low levels of A1 beta-casein [31].

The proportion of A1 to A2 beta-casein within bovine herds will depend on the relative frequency of the A1 and A2 alleles of the beta-casein gene sited on the sixth-chromosome [31]. The two alleles are co-dominant, meaning that a cow carrying one copy of each of the A1 and A2 alleles produces A1 and A2 beta-casein in equal amounts and is commonly termed an ‘A1A2’ cow. The relative frequency of the alleles varies between countries and between breeds, but typically the ratio at a country level in modern industries based on European breeds is between 1:3 and 3:1. Specific herds may lie outside these ratios depending on historical bull choices. Additionally, there are European niche breeds such as Guernsey and Fleckvieh which tend to carry higher levels of the A2 allele. There is also a tendency for breeds with Northern European origins to carry higher A1 levels than breeds with Southern European origins [31], with this flowing through to national herd data and consequent between-country differences.

Given that some cows carry one copy of each allele with associated co-dominance, the proportion of cows producing only A2 beta casein and termed ‘A2 cows’ will be less than the proportion of the A2 beta-casein in bulk milk in all situations where some A1A2 cows are present. At the retail level, milk in which all of the beta-casein is A2 is called ‘A2 milk’. In contrast, milk that contains some A1 beta-casein is commonly called ‘A1 milk’ despite typically also containing some A2 beta-casein.

There is also a human version of beta-casomorphin-7 with the structure tyrosine–proline–phenylalanine–valine–glutamine–proline–isoleucine, denoted here as hBCM7 (Figure 2). However, the human version of BCM7 is a much weaker opioid than the bovine form [32]. It is found mainly within colostrum and early lactation-stage milk [33] and it has been postulated to play a role in bonding of baby to mother. It is also much more susceptible than bBCM7 to internal cleavage by peptidases given the internal phenylalanine–valine–glutamine sequence. The specific mechanisms leading to hBCM7 in early-stage breast-milk remain to be fully elucidated. However, it is clear that hBCM7 is released in much smaller quantities than is the case with bBCM7 in most commercial milks, with this being entirely logical given that human beta-casein is of the A2 type.

In summary, it has become progressively evident over the last 40 years that the important casomorphin from a human-health perspective is bBCM7, with this peptide being preferentially released by cattle carrying beta-casein alleles belonging to the A1 family. Milk from cattle exclusively carrying double copies of the A2 allele, and also the milks from all other species of animal, are generally not considered to have casomorphin issues that are of public-health concern.

## 5. Casomorphin Effects on Human Health

### 5.1. Type 1 Diabetes

The initial evidence linking casomorphins to specific health issues was epidemiological with the investigational hypothesis deriving from clinical-practice insights. Between-country population studies identified remarkable associations between the intake of A1 beta-casein and the incidence of Type 1 diabetes in children with data from years 1990–1994 [34,35]. The hypothesis driving those investigations arose from empirical evidence that Polynesian children living in New Zealand had much higher levels of Type-1 diabetes than Polynesian children living in the Pacific Islands, with the key dietary difference being the much lower quantity of milk consumed in the Islands. The other evidence contributing to the initial hypothesis was that African children on very high-milk diets and in situations where the beta-casein was of the A2 type from indigenous African cattle were not susceptible to Type-1 diabetes [24,36]. The notion that milk might be a contributory causal factor for Type 1 diabetes was well established at that time [37,38,39] but the suggestion that the causal factor was a specific opioid-derived milk peptide was novel. Given the very high levels of statistical significance from the hypothesis-driven A1 versus A2 epidemiological investigations, the associations were unlikely to be random, and could not be argued to be a consequence of data-mining. However, a counter perspective was that between-country associations can never provide proof.

Given the widely accepted evidence that Type 1 diabetes is an auto-immune disease, then, if A1 beta-casein is a causal factor, it is logical that the antigen has to be the difference between the two beta-caseins in relation to the release of bBCM7. This logic is reinforced by evidence for a homologous peptide embedded within beta-cells within the pancreas that has the same sequence as the last four amino acids of bBCM7 [40]. Additionally, there is historical evidence from Germany that Type 1 diabetes sufferers have high levels of A1 beta-casein antibodies relative to those who do not have the disease [41]. However, at that time and ensuing times, there was no acknowledged mechanism by which bBCM7 was transported from the gut system to the pancreas. Accordingly, the mainstream medical perspective remained that bBCM7 would be broken down by the enzyme DPP4 before entering the circulatory system and was therefore unlikely to be relevant.

Subsequently, there has been a stream of work linking Type 1 diabetes to intestinal permeability and also to particular microbiota. Vaarala and colleagues in 2008 described the concept of a ‘perfect storm’ of aberrant intestinal microbiota, a leaky intestinal mucosal barrier, and altered intestinal immune responsiveness linking to other susceptibility factors [42]. Of relevance here is that the enzyme DPP4, which is the only peptidase enzyme known to break down bBCM7, is a brush-border enzyme, also present in serum, but human DPP4 does not circulate within the gut contents. Impaired brush border DPP4 production is clearly associated with intestinal permeability [43]. There is also a literature showing that specific microbiota themselves produce DPP4 and hence have the ability to degrade casomorphins and other food-derived opioids [44,45]. Additionally, there is a separate stream of work that now demonstrates that bBCM7 is present both in blood and urine [46,47]. Accordingly, some key elements of the initial skepticism have now been invalidated.

A more recent paper has developed a more sophisticated argument that A1 beta-casein and hence bBCM7 is a primary trigger for Type 1 diabetes while recognizing, consistent with the above knowledge, that there is a multiplicity of influencing factors that can enhance the opportunity for bBCM7 to act as the trigger [48]. There is also evidence of epigenetic effects [49]. It is also possible for bBCM7 to be a powerful mediating factor without necessarily being the final trigger. For example, the final trigger may include viruses, impacting via intestinal damage and decreased brush-border DPP4 biosynthesis [50]. Accordingly, it is not helpful to develop arguments based on single-factor causation expressed on an ‘either/or’ basis.

Within a broader framework, any factor that increases intestinal permeability by damaging the mucosal barrier is likely to increase susceptibility. It is notable that bBCM7 stimulates mucin biosynthesis [51,52,53], with this likely to be a protective response. It is also notable that although genetic susceptibility to Type 1 diabetes and the role of gut permeability are well established, direct linkage of Type 1 diabetes to specific microbiota and dysbiosis is relatively new and of increasing interest [54,55].

One issue that has not previously been integrated within the overall casomorphin and Type 1 diabetes synthesis is the role of opioid receptors within the pancreas and Islets of Langerhans where the insulin-producing beta cells are located. However, it has been known since the late 1970s that both mu-opioid and delta-opioid receptors are located there [56,57]. More recently, an extensive literature has been developing linking mu-opioids in the brain to insulin secretion [58,59]. This provides evidential logic as to why bBCM7 will be attracted to the pancreas and how it can therefore be expected to interfere with endorphin messaging. Given the homology of bBCM7 to sequences within the GLUT2 molecule, together with cross-reactivity of beta-casein T cell lines to human insulinoma extracts and GLUT2 peptide [60], there is a credible pathway as to how an autoimmune reaction then occurs, leading to auto-immune destruction.

### 5.2. Heart Disease

As with Type 1 diabetes, major differences in incidence between countries within the developed world correlate remarkably with A1 beta-casein intake [34,61]. Supporting evidence is provided by a causal relationship identified in rabbits having been exposed to A1 beta-casein intake versus A2 beta-casein intake, leading to increased deposits of arterial plaque with A1 beta-casein [62]. Additionally, formula-fed babies have been shown to have high antibodies to oxidized LDL [63], with this being associated with A1 beta-casein intake [64]. In piglets, a direct trial comparison between A1 beta-casein and A2 beta-casein intake has shown a statistically significant difference in relation to oxidized LDL antibodies, but it is only published in the Czech language [65]. It had previously been demonstrated that BCM7 has the ability to catalyze the oxidation of LDL [66]. In humans, it has long been recognized that high milk diets for stomach ulcer sufferers (a potential cause of stomach permeability) lead to high death rates from heart disease [67], and there is hypothesis potential for this linking to A1 beta-casein with release of bBCM7 within sera following the absorption of larger protein components.

The same logical pathway linking the passage of bBCM7 from the gut to the circulatory system occurs for heart disease as for Type 1 diabetes. There is also extensive evidence of the presence of mu, delta, and kappa opioid receptors within the cardiovascular system [68]. However, although the presence of opioids within heart muscles has been known for a considerable time [69], there is no evidence of this knowledge of opioids and opioid receptors being previously linked to risk issues associated with food-derived opioids. One difference between heart disease and Type 1 diabetes is that whereas Type 1 diabetes susceptibility is clearly associated with specific HLA haplotypes, any such links remain highly speculative in relation to heart disease apart from rheumatic fever, which is known to have genetic HLA factors [70].

### 5.3. Links to the Brain

There is extensive evidence that bBCM7 crosses the blood–brain barrier and affects behavior and physiology in multiple ways and that this is carrier-facilitated [71]. As one example, for more than 30 years bBCM7 has been suspected as a cause of sudden infant death syndrome (SIDS) following evidence that bBCM7 induced apnea and irregular breathing in both adult rats and newborn rabbits [72]. Subsequently, casomorphins were identified in the brainstems of children who have died from SIDS [73] but comparisons with normal children are obviously not possible. This evidence was subsequently integrated by Cade and colleagues in 2003 [71]. Thereafter, there was a research hiatus in relation to respiratory depression until evidence from Poland was published showing that babies who suffer acute life-threatening events (ALTE) through apnea are characterized by circulating levels of bovine BCM7 that are three times higher than in normal children [47]. These same ALTE children had DPP4 serum activity levels only 58 ± 3% of those in normal children.

Russian scientists have found that bovine BCM7 enters the blood of babies fed milk-formula diets [74]. Whereas some of these babies could quickly metabolize the BCM7, others were slow metabolizers. It was found that where bovine BCM7 levels in the blood stayed high between feeds, there was a high risk of delayed psychomotor development. The cause of slow metabolization was not explored or discussed within that specific research, but insufficient ability to upregulate DPP4 must be considered as a prime explanatory contributor.

Bovine BCM7 has long been considered a risk factor for autism, but the hypothesis remains controversial. Cade and his team integrated evidence linking autism and schizophrenia to casein and gliadin and characterized the conditions as intestinal disorders [75]. Trials with animals show that when bBCM7 crosses the blood–brain barrier, it leads to autistic type behavior [76]. Milk elimination trials in humans have produced positive results [77,78] but are often criticized for lack of double-blind protocols. Many autistic children suffer from digestive complaints, which may make them susceptible to bBCM7 absorption [75]. Recent research confirms bBCM7 in serum is associated with upregulation in serum of DPP4 [79]. It is also known that bBCM7 reaching the brain affects the serotonergic system [14] with potential implications for neurological development.

More recent research has extended neurological evidence in otherwise healthy people from behavior to cognition in adults [80] and children [81]. It has been found that the consumption of milk containing A1 beta-casein decreases both cognitive processing speed and accuracy. The key difference between behavior and cognition measures is that behavior is a response to stimuli, whereas cognition relates to information processing.

Trivedi and colleagues have demonstrated across a range of both in vitro [82,83,84] and in vivo investigations [27] that A1 beta-casein and bBCM7 are inflammatory, including decreased glutathione and cysteine expression in a range of epithelial and neuronal cells together with decreased DNA methylation in differentiating neural stem cells, with redox and epigenetic implications. They also report major divergence between bBCM7 and hBCM7 in redox status and neurogenesis, consistent with a hypothesis that hBCM7 plays a positive role in neurogenesis contrasting to the negative role of bBCM7. From these overall findings, they draw links to a body of evidence demonstrating that the same glutathione, methylation, redox, and inflammatory parameters are also associated with development of autistic conditions and Alzheimer’s disease.

### 5.4. Gut Conditions

The evidence linking A1 beta-casein to specific digestive issues has developed almost exclusively since 2010. When the first edition of my book ‘Devil in the Milk’ was written in 2007 [36] covering various health conditions relating to A1 beta-casein, it was not possible to present any significant human or animal scientific evidence relating to digestive intolerances or gut inflammation that came from trials incorporating treatments plus comparative controls. Instead, the evidence was restricted to anecdotal case information. However, since then, an extensive literature has developed [22], initially with rodents and subsequently focusing on human trials. A paper published in 2013 found that consumption of A1 beta-casein by mice induced inflammatory responses in the gut by activating a Th2 pathway as compared to A2 beta-casein [12]. Significant differences included myeloperoxidase (MPO) activity, inflammatory cytokines, various antibodies including IgE and IgG, and mRNA expression for toll-like receptors (TL2 and TLR4). A paper published in 2014 found that consumption by Wistar rats of milks containing A1 beta-casein compared to A2 beta-casein delayed intestinal transit and increased inflammatory status measured by MPO activity, with both of these being negated by pre-treatment with naloxone, confirming that the effects were opioid-related [28]. In the same trial, DPP4 activity was also upregulated in the jejunum by the milk containing A1 beta-casein but this was identified to be by a non-opioid mechanism as it occurred independently of naloxone treatment.

The first human trial comparing the gastrointestinal effects of milks containing only A1 beta-casein in comparison with milks containing only A2 beta-casein, undertaken as a cross-over trial in Australia and published in 2014, found higher Bristol Stool readings with A1 beta-casein compared to A2 and that gut pain was strongly correlated with higher Bristol stool readings when participants were on the A1 diet [85]. Then, in 2015, a much more detailed crossover-design investigation in China, comparing milk containing a mix of A1 and A2 beta-casein (A1A2) with milk containing only A2 beta-casein (A2A2), found that the A1A2 milk resulted in significantly greater digestive discomfort, higher concentrations of inflammation-related biomarkers, longer gastrointestinal transit times by on average approximately 6 h, and lower levels of short-chain fatty acids [86]. The digestive discomfort symptoms on the A1A2 milk relative to baseline (after dairy washout) increased in both lactose-tolerant and lactose-intolerant subjects, with lactose status identified from urinary galactose, whereas when on the A2A2 milk, these symptoms did not increase relative to baseline for either lactose-tolerant or lactose-intolerant subjects. The authors concluded, inter alia, that given the relative benefits of the A2A2 milk for both lactose-tolerant and -intolerant subjects, some perceived symptoms of lactose intolerance may stem from inflammation relating to A1 beta-casein and its derivative bBCM7. Two further Chinese crossover studies, one of 600 adults [87] and another of 80 school children [81], have confirmed the previous digestive discomfort findings relating to A1 versus A2 beta-casein and also provided confirmatory evidence supporting interactions between A1 beta-casein and lactose intolerance that did not apply with the A2 beta-casein diet. A subsequent American study published in 2020 has confirmed the findings that discomfort symptoms among persons who consider themselves lactose intolerant are increased when the milk contains A1 beta-casein [87]. Additionally, in a subset of subjects who were identified as lactose maldigesters using a hydrogen breath test, the level of breath-hydrogen was higher when the diet contained A1 beta-casein, despite the milks containing no difference in lactose content. Accordingly, in explaining the consistent evidence for interactions between A1 beta-casein and lactose intolerance symptoms, there would seem to be at least two logical deductions from the evidence that are worthy of consideration. First, the delay in intestinal transit creates opportunities for enhanced fermentation of lactose that has not been digested by lactase. Second, inflammation associated with A1 beta-casein may reduce the residual ability to produce the lactase enzyme.

An area requiring further study includes differences in casein micelle structure and reduced chaperone ability in the gut of A1 versus A2 beta-casein identified in Australia by Raynes and colleagues [88]. The authors report that chaperone functionality is important for reducing aggregation of other proteins including whey proteins, with protein aggregation of potential relevance not only within the gut itself but linking through to a range of neurological decay conditions.

### 5.5. Other Conditions

A1 beta-casein has recently been implicated as a predisposing factor for asthma [89]. It has also been hypothesized that bBCM7 may influence fractures and obesity via mu-opioid pathways [90]. Milk has also been linked by epidemiology to multiple autoimmune conditions, including both Parkinson’s [91] and multiple sclerosis [92,93]. However, specific causal agents such as bBCM7 are difficult to evaluate within either epidemiological or clinical settings.

The presence of opioid receptors had been identified by the late 1990s in the adrenal glands, kidney, lung, spleen, testis, ovary, and uterus [19]. Opioid receptors and opioid effects have also been evident since the 1990s in relation to the immune system [94], although aspects thereof remain to be elucidated [95]. The role of opioids and opioid receptors is also explicit in relation the endocrine system [96,97]. Mu-opioid receptors and opioid sensitivity are also associated with hepatic conditions [98,99]. Accordingly, with the opioid characteristics of bBCM7 well-established, together with comprehensive evidence of widespread presence of opioid receptors throughout the human body, there is a logical underpinning to a set of intriguing hypotheses for investigations across a broad range of inflammatory and autoimmune diseases.

Conversely, no published research has been found suggestive of any benefits of bBCM7 as a dietary component. However, there is a set of papers linked to Nanjing Agricultural University exploring the use of BCM7 as a drug to reduce the effects on various health parameters in rats with diabetes induced artificially by administration of streptozotocin [100,101,102].

## 6. Gliadin Peptides and Gliadorphin

The key gliadin peptide of opioid significance and where issues align with the casomorphins is gliadorphin-7, also often termed gliadomorphin and gluteomorphin. The amino-acid structure is tyrosine–proline–glutamine–proline–glutamine–proline–phenylalanine, and for simplicity this peptide is hereafter termed GD7. Key features in common between GD7 and bBCM7 are the Tyr-Pro at the N-terminus and two further proline amino acids in close succession thereafter (Figure 3). These features, as with bBCM7, will ensure opioid characteristics plus resilience to degradation of GD7 from the C-terminus. However, homology only extends to four of the seven amino acids in total, leading to the potential for considerable differences in specific opioid characteristics. As with bBCM7, shorter peptides deriving from GD7 are unlikely to be of importance within natural in vivo settings given the resilience to degradation from the C-terminus. However, given the diversity of gliadin proteins within the Triticeae, there is potential for other specific gliadorphin structures to be identified.

In many respects, the investigatory situation with GD7 is considerably more complex and confusing than is the case with bBCM7. As a starting point, coeliac disease is associated with gluten in a much broader context than just GD7 [10]. Given the acute nature of coeliac disease, it has therefore inevitably overshadowed the study of more chronic and in some cases delayed conditions arising from GD7. Any trial removing gluten from the diet has inevitable confounding factors. These include issues such as fermentable short-chain carbohydrates (FODMAPs) [103] and glyphosate [104]. Additionally, the blinding of participants in GD7 investigations is scarcely possible given the need to remove all wheat, barley, and rye from the diet. In contrast, it has been possible to conduct blinded trials incorporating treatment and control of A1 versus A2 beta-casein without subjects automatically identifying the alternative diets and also without confounding from other variables.

Despite the dominance of coeliac research compared to non-coeliac gluten-intolerance research, there is evidence of serological differences between these two conditions, with IgG gliadin antibodies common but not always present in both groups, whereas IgG deamidated gliadin antibodies, IgA transglutaminase antibodies and endomysial antibodies are all typically found in coeliac subjects but not in non-coeliac intolerance subjects [105]. This highlights the fundamental difference between the two conditions.

An alternative investigational approach is therefore to draw on insights from casomorphin research, particularly in relation to bBCM7, and question whether there is comparable evidence for GD7 independent of a specific range of gluten-protein issues associated with coeliac disease.

A starting point is to recognize that in situations where bBCM7 is able to enter the circulatory system, relating particularly to intestinal permeability and in situations of decreased upregulation-potential for the brush-border DPP4, then the same conditions are likely to exist for GD7. Similarly, to the extent that GD7 is also a mu-opioid, or alternatively a delta-opioid, then it is likely to be attracted to the same organs as bBCM7, although not necessarily the same opioid receptors [106]. Different mechanisms by which these two peptides pass the blood/brain barrier have been identified, with Sun and Cade identifying that GD7 passage to the brain was restricted to diffusion through circumventricular organs while bBCM7 passes the BBB by carrier facilitation [71].

In relation to Type 1 diabetes, both milk and cereal diets have been implicated. For example, the non-obese diabetic (NOD) mouse strain was specifically bred as susceptible to cereal diets. The debate between A1 beta-casein and cereal diets as alternative triggers has at times been controversial [107,108]. However, once BCM7 and GD7 are considered as potential ‘partners in crime’ based on their peptide homology, combined with the accepted understanding that Type 1 diabetes is an autoimmune disease, then a different perspective is created.

In relation to psychological issues, there have long been theories that schizophrenia is related to cereals. Among other susceptibility factors, coeliac disease will create the underlying gut environment facilitating passage of high-proline peptides through the gut barrier. Accordingly, the neurological issues associated with coeliac disease and non-coeliac gluten intolerance such as ‘brain fog’ [109] are consistent with the specific peptides being either one or both of GD7 and bBCM7. It is also notable that a combined gluten-free and casein-free diet has been widely identified as relevant to autistic attributes. Similarly, when bBCM7 and GD7 were infused into rats, then both induced immunoreactivity in geniculate nuclei and the alveus hippocampus in a dose-related fashion, albeit with much higher doses required for GD7 than bBCM7, and with all effects prevented or reduced by naloxone treatment [76,106]. However, whereas bBCM7 elicited “bizarre behaviors”, this did not occur with GD7 [106].

There is also an emerging literature linking gliadin antibodies to rheumatoid arthritis, with many and perhaps most of these cases not related to coeliac disease [110,111].

Evidence from Trivedi and colleagues [82] has demonstrated the commonality of effects of GD7 and bBCM7 decreasing cysteine uptake in cultured human neuronal and gastrointestinal epithelial cells via the activation of opioid receptors, albeit with bBCM7 effects being stronger. However, given that the major focus of their work was on casomorphins, there is considerable scope for further investigation of these issues with GD7. As with bBCM7, it is evident that the role of GD7 is worthy of consideration across the spectrum of autoimmune conditions.

There is no direct evidence that GD7 is a beneficial peptide for human health. However, it is part of the complex of gluten proteins that have important properties relating to the texture and form of cereal products. The importance of GD7 specifically to these attributes is not clear.

## 7. Links to Microbiota

There is a fast-developing literature in relation to microbiota and the gut–brain axis, with comprehensive recent reviews from Mayer et al. [112] and Martin et al. [113]. However, major questions remain to be answered as to the mechanisms by which crosstalk between the gut and brain might occur and also how that crosstalk is disrupted. Current evidence suggests that multiple mechanisms, including endocrine and neurocrine pathways, may be involved in gut microbiota-to-brain signaling and that the brain can in turn alter microbial composition and behavior via the autonomic nervous system [112]. Gut to brain connections are also linked to short-chain fatty acids [114].

In contrast to the generic gut–brain axis literature, the literature relating to both opioid drugs and food-derived opioids as modulators and disruptors of the gut–brain system is at a much earlier stage of development, but with a recent contribution from Spanish researchers representing a significant synthesis [115]. It is evident that food-derived opioids have the potential to be a generic and chronic source of system disruption in otherwise healthy populations. Within this framework, there is potential for dysbiosis and inflammation to be cross-influencing factors, but with food-derived opioids as the external source of system disruption.

Links between microbiota and food-derived opioids independent of the gut–brain axis are strongly evidential, albeit with the need for further investigations of both effects and specific mechanisms. To date, this existing evidence is not well-integrated into the gut–brain literature. Indeed, it is notable that the generic gut–brain axis literature has not placed a greater focus on external disruptors to the system. There may also be a need to take a broader system-based human biology perspective incorporating all internal organs and peripheral tissues that contain opioid receptors, given the fundamental messaging role of these receptors.

Differences in short-chain fatty-acid production between conventional milk containing an A1/A2 mix versus A2 milk with all beta-casein of the A2 variant were recorded in two blinded crossover Chinese trials [80,81], with considerably higher levels of short-chain fatty-acids, specifically butyrate, acetate, and propionic, when all beta-casein was A2, and with all differences to the A1-containing milk at very high levels of statistical significance (*p* < 0.001). The relevance of these results is that short-chain fatty-acids are produced by bacterial fermentation within the colon, and these fatty acids are therefore a proxy for specific microbial activity.

A recent Italian study [116] of diets containing milk from A2A2 cows versus milk containing a mix of A1 and A2 beta-casein provided to ageing mice identified consequential differences in microbiota, with A2 milk leading to higher levels of short-chain fatty acids including butyrate. This affected the gut immunological phenotype and favored CD4^+^ T cell differentiation, resulting in improved gut villi morphology. The authors reported that this was the first known investigation of these issues within an ageing model. They concluded that A2A2 milk type may be suggested as a suitable strategy to achieve positive gut health outcomes in ageing populations.

A Danish study [117] found evidence for a low-gluten diet inducing changes in the intestinal microbiome of healthy Danish adults but the interpretation thereof is more complex given that these results can be confounded by FODMAP issues in wheat diets [103]. However, a recent study from Germany [118] that sought to distinguish between FODMAP and gluten issues in cereal diets concluded that both issues were relevant to non-coeliac gluten disorders, with gluten-free diets contributing independently of FODMAP issues. A recent Spanish study [119] found that a low-gliadin transgenic wheat produced a preferred microbe profile with higher butyrate-producing bacteria compared to a cereal-free gluten-free diet, with implications for reduced gut permeability.

A recent Italian review [120] concluded that for persons not exhibiting gluten sensitivity, a gluten-free diet can cause depletion of beneficial species, e.g., Bifidobacteria, in favor of opportunistic pathogens, e.g., Enterobacteriaceae and Escherichia coli, whereas in both coeliac and non-coeliac gluten sensitivity conditions, a gluten-free diet evokes a positive effect on gastrointestinal symptoms by helping to restore the microbiota population and by lowering pro-inflammatory species. A recent Mexican study [121] provides supporting evidence through the upregulation of *Pseudomonas* species in duodenal biopsies of patients with both coeliac and non-coeliac intolerances disorders, but particularly in non-coeliac gluten-intolerant patients, noting that *Pseudomonas* comprises strains with gluten-degrading capabilities. From an ecological perspective, contrasting microbiota evidence on a gluten-free diet between people who do and do not normally experience gluten sensitivity is not necessarily surprising, with specific gluten-degrading microbiota losing their competitive advantage in the absence of these opioid peptides within the diet.

There is increasing evidence that some microbiota have bacterial equivalents to human DPP4 [44,122], and this may include yogurt bacteria [123], with this creating the potential for attenuation of opioid peptides passing from the intestines to the circulatory system in individuals with a compromised intestinal barrier.

A recent Danish study [124] investigated whether gliadin would impact the metabolic effects of an obesogenic diet using a mouse model. Mice fed the gliadin-added high-fat diet displayed higher glycated hemoglobin and higher insulin resistance, more hepatic lipid accumulation, and smaller adipocytes than mice fed the gliadin-free high-fat diet. This was accompanied by alterations in the composition and activity of gut microbiota, gut barrier function, urine metabolome, and immune phenotypes within liver and adipose tissue. The authors concluded that gliadin disturbs the intestinal environment and affects metabolic homeostasis in obese mice, suggesting a detrimental effect of gliadin, and hence gluten intake, in normally gluten-tolerant subjects consuming a high-fat diet.

Beyond the specifics of the gut–brain axis, there is increasing evidence of the microbiome being relevant to a range of conditions. For example, the role of the microbiome in relation to Type 1 diabetes has recently become an increasing field of investigation [44,45,54,55], following earlier work implicating gut permeability as a key factor [42]. Similarly, there is detailed evidence that the microbiome is associated with cardiovascular conditions [125]. Given the almost ubiquitous presence of casomorphins and gliadorphins within diets, the question of whether the microbiome is a causal factor in these conditions independent of the food-derived opioids is unresolved. This can only be solved by bringing food-derived opioids, the microbiome, and these other conditions into an integrated research program.

More generally, the interpretive challenge with microbiota arises from the complexity of associations between the food-derived opioids, the short-chain fatty acids, the presence of inflammatory markers, the neurological effects of behavior and cognition, the complexity of crosstalk between the brain and the gut, and effects on many other organs that have opioid receptors. Within this overall complexity, the food-derived opioids are clearly an external flow into the system, with other factors such as microbiota having potential to be either directly or indirectly consequential and setting up cascading sequences of effects that then modulate and reinforce other specific outcomes. Whereas the knowledge of the role of the gut within food-derived opioid conditions can be considered as extensive albeit incomplete, the explicit role of specific microbiota, together with the molecular mechanisms and impacts of food-derived opioids within both the gut–brain axis and the broader human biological system, remain medical frontiers.

## 8. Conclusions

There is a broad range of evidential material linking food-derived opioids with delayed intestinal transit, intestinal inflammation, intestinal permeability, and an altered microbiome, with this being linked via opioid receptors and some other receptors to conditions affecting many organs. The diverse presence of opioid receptors within human tissue, combined with individual genetic differences, provides an explanation as to why the effects can be diverse. This then links to inflammatory and auto-immune outcomes in those organs. What remains to be elucidated is the precise nature of the interactions and influencing factors. Many auto-immune relationships remain speculative as to cause.

Whereas many of the short-term gut effects of bBCM7 can be investigated within clinical settings of double-blind random treatment-and-control investigations, this is more challenging with GD7 because of the need to isolate GD7 from other protein and non-protein components of cereals. In relation to the role of the microbiome, elucidation of the role is difficult given the complexity of the system containing direct and indirect effects, multiple interactions, and feedback loops. What is apparent is that the microbiome provides gut-condition indicators and gut-condition modulators, with this occurring in association with other factors and hence is occurring in a systemic framework. The biosynthesis of human DPP4 within the brush-border of the gut system, combined with bacterial DPP4 linked to the presence of particular bacteria within the microbiome, together with ability to upregulate DPP4 within the sera and other organs, are all likely to be of fundamental importance. When the microbiome is considered an ecological system, which itself is encompassed within a broader human system spanning the gut, brain, and internal organs with which it interacts, then a key question is what are the externally sourced causal modulators and disruptors of that system? It is in that context that more attention needs to be given to the food-derived opioids.

Specific strategies for reducing exposure to bBCM7 and GD7 are two-fold. Bovine BCM7 is relatively easy to remove from the food-system by producing cows that produce A2 rather than A1 beta-casein or alternatively placing more emphasis on milks from other species such as sheep and goats, plus an emphasis on human milk for babies. Removing GD7 from the diet is more difficult because currently it requires removing gluten and hence all cereals containing gluten from the diet. However, technical solutions such as genetic manipulation to alter one or two amino acids within the GD7 sequence may in the future become practical.

## Figures and Tables

**Figure 1 ijerph-18-07911-f001:**
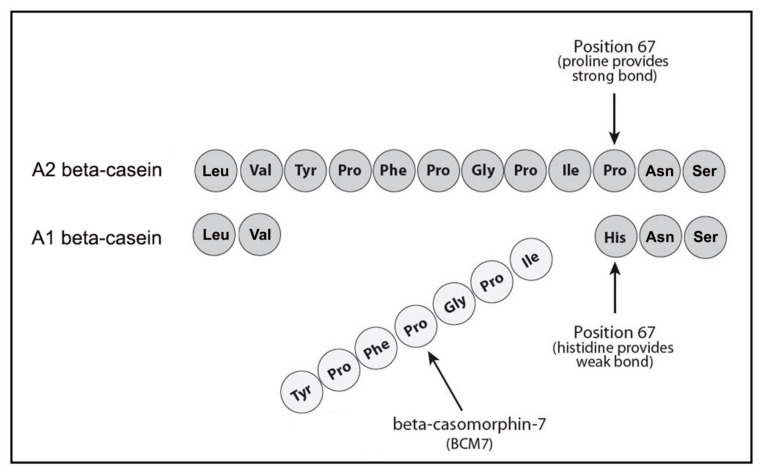
Preferential release of bovine beta-casomorphin-7 from A1 beta-casein.

**Figure 2 ijerph-18-07911-f002:**
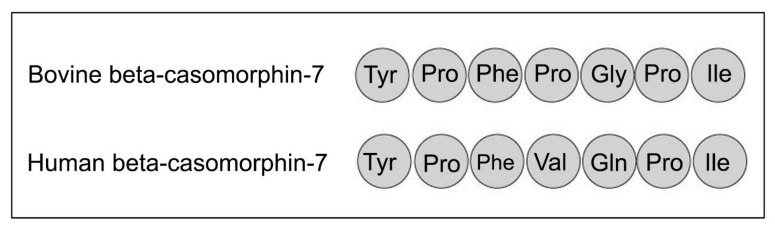
Comparing human and bovine beta-casomorphin-7.

**Figure 3 ijerph-18-07911-f003:**
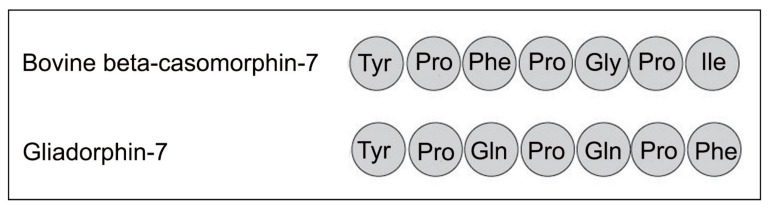
Comparing bovine beta-casomorphin-7 and gliadorphin-7.

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
