# Peer review of "Casomorphins and Gliadorphins Have Diverse Systemic Effects Spanning Gut, Brain and Internal Organs"

_ijerph, 2021, doi:10.3390/ijerph18157911_

Round 1

Reviewer 1 Report

For a start, I would like to emphasize that I consider myself lucky for giving me the opportunity to review an article that is being negotiated with such important issues as “Food-derived opioid peptides”, especially Casomorphins and gliadorphins.

My comments on the manuscript are as follows:

  1. Title: In the “main body” of the article exist sections that analyze correlations with Type 1 diabetes and Heart disease. So why only “from gut to brain”? The title should be adequate to the content of the article. Please consider to revise it.
  2. The entire manuscript is very "heavy" and unattractive, so the insertion of figure(s), diagrams or tables makes it easier to read and with more objective information.
  1. Lines 108-109: Please give examples from the literature.
  2. Lines 112-165: It would be better to place the data in tables or figures.
  3. Lines 166-175: It is human milk “food” in the broadest sense of the word. Because it concerns a specific age group. Also, several epidemiological studies support the protective role of breastfeeding in reducing the risk for type 1 diabetes. Please author to respond to my attention and if there is evidence concerning human milk, Casomorphins and public health.
  4. Lines 183-271: Despite a number of reports demonstrating a correlation between dairy consumption (casein) and type 1 diabetes, heart disease,  conclusive evidence for Casomorphins (bBCM7)  is lacking. It should be considered that casein automatically gives Casomorphins?
  5. Section “Links to the Brain”: What can we say about autism? In autism, inhibition of alimentary enzyme results in breakdown of casein to casomorphin, instead of to amino acids (Yagil, 2004). Casomorphin is a more potent opioid than morphine itself and responsible for most of the typical behavioral and cognitive symptoms of autism.
  6. Section “Gliadin peptides and Gliadorphin”: This topic occupies a small part of the whole article, where there is no link to public health issues.
  7. Please discuss the methodological limitations of the article.

Author Response

Thank you for your constructive comments, which have been very helpful in strengthening the paper

In relation to your specific points.

First 2 comments:

  1. I have revised the title as suggested
  2. I have entered three diagrams

Second set of comments

  1. I have reworked this material

  2. As above

  3. I am cautious of getting into the debate here about breast feeding. I have highlighted the differences between bBCM7 and hBCM7 including a diagram.

  4. Yes, casein will always produce casomorphins but not necessarily BCM7. I have extended the information in relation to BCM9. Throughout this paper I have been careful to use terms such as ‘evidential’, ‘associations’ and ‘links’ and I have not asserted ‘proof’. I have restricted statements of causality to situations where there is a body of statistically significant clinical evidence from treatment comparisons.

  5. I searched and found a reference by Yagil 2004 (http://camelnet.eu/wp-content/uploads/2014/09/Camel-milk-autoimmunity.pdf) focusing on camel milk and its characteristics. Yagil says that camel milk contains no beta-casein but I think that is unlikely. Rather, my understanding of the milk of camels is that it contains beta-casein that is of the A2 family, (as are all non-bovine species), and the thrust of the Yagil paper, which essentially is observational, is consistent with the perspectives of my paper. The Yagil perspective on ‘casomorphin’ is limited by non-recognition of the diversity of casomorphins. In relation to autism, I have chosen to use more recent papers.  I believe I have covered the current diversity of scientific opinion in relation to autism, including noting that the benefits of both GFCF diets and the suitability of A2 beta- casein in regard to autism are evidential but remain controversial. Given that my focus is integrative across the human system I have had to be selective with choice of references consistent with acknowledging diversity of scientific thinking.

  6. The material in my paper on gliadorphins is less detailed than for casomorphins but that aligns with less research having been undertaken. I have now emphasised that within the paper. When originally framing this paper, I gave consideration as to whether to limit it to casomorphins, but I decided that the current evidence plus the homology between bBCM7 and GD7 (which I have now highlighted through Figure 3), combined with the importance of both dairy and gluten-containing cereals in modern diets, merited broadening the treatment to include gliadorphin-7 in some depth, albeit constrained by this being one of science’s frontiers. I cannot agree that there is no link to public health, with explicit linkages to conditions of the gut, pancreas and brain, and with the logic presented to support additional hypotheses worthy of investigation. I have reworked some of the material in the section on gliadorphins to provide more clarity on these points.

  7. I have added a small section explicating the methods used for literature searching and filtering thereof within the specific focus of the manuscript.

Reviewer 2 Report

The manuscript reviews the effects of two peptides derived from milk and gluten, such as casomorphins and gliadorphins. The document is well presented, well structured, with enough recent and reliable references, and contains few "typos," so that except for a few suggestions and minor concerns, I would be in a position to recommend its acceptance.
a) A figure comparing the structures of GD7, bBCM7 and bBCM9 would greatly help the reader to appreciate the structural differences. Although the sequences are mentioned in the text, the inclusion of an image would allow a better understanding of the author's analysis.
b) A figure summarizing the effects found for both types of peptides would also make the manuscript easier to read. Mainly, there are effects caused by both peptides and other effects that have only been evidenced in BCM7 and only in GD7. Maybe a Venn diagram would be valid?
c) Are there studies on the bioavailability of both peptides?
d) The reviewer considers that it is necessary to expand the section related to the microbiota since, as the author mentions, it plays a significant role in the permeability and metabolism of peptides
e) Are there any beneficial health effects that are reported for BCM7 and GD7? Although it is understood that the review is mainly focused on their adverse effects, the reviewer considers that it would be necessary to mention their potential positive impacts as well.

Author Response

Thank you for your supportive and constructive comments.

In response:

a) I have added three diagrams to assist with communicating the structural differences between the peptides.

b) I have given careful consideration to this suggestion. I think the complexity is such that a Venn diagram might mislead. However, I have reworked material to provide greater clarity. Also, I have added a new section on the role of opioid receptors at an early stage of the paper.

c) There is evidence within the paper of bioavailability of BCM7 within the gut system, sera, brain and excreted in urine. Whether the BCM7 is bio-available in particular organs will depend on a multiplicity of factors including intestinal permeability, ability to upregulate DPP4 and genetic factors such as HLA status. BCM9 is clearly bio-available in the gut (now linked in the paper to ref 22, a systematic review of which I am a co-author). There is evidence reported in the paper for hypertensive benefits (ref 25) and antioxidant effects (ref 26) but I cannot find any research evidence of passage of the bBCM9 peptide itself beyond the gut system.

d) I have searched the literature again and have added multiple additional references within the microbiota section, albeit constrained by this being a science frontier, particularly in relation to food-derived opioids and the gut-brain axis.

e) I have now made explicit mention that no beneficial benefits of either bBCM7 or GD7 as part of a normal diet have been identified. I have also now noted research on bBCM7 as a drug treatment for streptozotocin-induced Type 1 diabetes in rodents.

Reviewer 3 Report

The manuscript titled: “Casomorphins and gliadorphins have diverse systemic effects spanning from gut to brain” presents interesting systemic effects from these food-derived opioids. The review is very well written and scientifically sound, I would only add some minor comments to be suitable for publication in IJERPH.

ABSTRACT

  1. Conclusions provided do not seem to have a relationship with the main purpose or topic exposed here.

MAIN TEXT

  1. Lines 30, 32, and 38 contains “influence” or “influencing”. Please replace with a suitable synonym.
  2. Please place keywords next to Abstract.
  3. I believe this manuscript will benefit with tables summarizing all the health effects of these opioids, as the document do not contain any figure or table. Please add a table of the main effects. Moreover, I suggest the author to add a figure summarizing the main mechanistic pathways at cellular or tissue level in which food-derived opioids act.

Author Response

Thank you for your supportive comments and constructive advice.

  1. I have reworked the abstract and conclusions for alignment. It has been challenging within the 200-word limit to say all that I would have liked within the Abstract, but your comment has been helpful.
  2. I have now used synonyms in two of these places
  3. Yes, that was in error that the keywords were also accidentally placed after the introduction
  4. I have added three diagrams. I have also considered adding a table of main effects. However, I soon found that a table became unwieldy given the complexities and systemic interactions. Accordingly, I have added a new section early in the paper on the role of opioid receptors and I believe this adds to the overarching structure and hence clarity of the paper. I have also tried to provide more clarity on these complex issues in multiple places within the paper.
    The question of mechanistic pathways at cellular level is complex, given the diversity of direct biochemical pathways, combined with endorphin blocking, combined with immune and auto-immune reactions. To provide further explication of pathways and associated insights, and to increase overall rigour and clarity, I have added approximately 20 references across the paper.  I have also given increasing acknowledgement that these issues are at the frontiers of knowledge, with much more to be learned.

Round 2

Reviewer 1 Report

Congratulations on your work.